# Active or Passive Commuter? Discrepancies in Cut-off Criteria among Adolescents

**DOI:** 10.3390/ijerph16203796

**Published:** 2019-10-09

**Authors:** Javier Zaragoza, Ana Corral, Sergio Estrada, Ángel Abós, Alberto Aibar

**Affiliations:** 1CAPAS-City (Centre for the Promotion of PA and Health), University of Zaragoza, 22001 Huesca, Spain; zaragoza@unizar.es (J.Z.); estradaten@unizar.es (S.E.); Aibar@unizar.es (A.A.); 2Faculty of Human Sciences and Education, University of Zaragoza, 22003 Huesca, Spain; 3Faculty of Health and Sport Sciences, University of Zaragoza, Huesca, 22001 Spain; aabosc@unizar.es

**Keywords:** active commuting, adolescents, cut-off criteria, distance

## Abstract

Active commuting to school has health implications for young people. Previous research has shown the need to consistently define the concept of “active commuter”, given that assessment as well as comparison between studies may be hindered by current discrepancies in frequency criteria. Using a sample of 158 Spanish students (12th–13th grade, 60.8% girls), the current study aimed to compare several cut-off criteria to rigorously identify the frequency of weekly active trips to school in order to categorize adolescents as active or passive commuters, and to analyze whether the threshold living distance to school is associated with the different trip cut-off criteria. Percentages of active commuters ranged from 75% to 88.6%, varying significantly depending on the cut-off criteria (5–10 active trips/week) used. The results also support the need to be stricter in the selection of a cut-off criterion when the distance to the school becomes shorter. Our findings highlight the importance of following a standard criterion to classify individuals as active or passive commuters, considering the characteristics of the context in which each study is conducted.

## 1. Introduction

Physical inactivity is the fourth leading risk factor of mortality worldwide [1]. The benefits of physical activity (PA) in adolescents are significant and widely accepted [2,3]. However, a vast body of international [4] and Spanish studies [5] showed that most adolescents do not meet international PA recommendations (i.e., 60 min of daily moderate-to-vigorous PA). Therefore, promoting PA in adolescents seems of paramount importance, not only to enhance current health benefits, but also to transfer them into adulthood [6,7].

Active commuting can be a promising strategy to promote PA among adolescents [8], providing an alternative to more traditional PA domains such as sport and exercise [9]. A large body of research has indicated that active commuting plays an important role in increasing the overall PA level in children and youth [10]. Promoting PA through walking and cycling is a priority of the World Health Organization, as reflected in the Global Action Plan on Physical Activity 2018–2030 [11].

Active commuting (i.e., walking or cycling) is an accessible and inexpensive kind of PA which can be easily integrated into adolescents’ daily routines. Although independent mobility increases in adolescence [12], the prevalence of active commuting to school (ACS) has markedly decreased in recent decades in many developed countries [13,14,15], and in the last decade in Spain [16]. However, comparisons are often complicated due to measurement differences. The literature has also shown some inconsistencies in the categorization of active and passive commuters. The heterogeneity of research questions and methods makes it difficult to compare the results and percentages of ACS between studies.

The most commonly asked question refers to a period called ‘usually’, more specifically, ‘How do you usually go to school?’ [17,18,19]. Most studies have reported the mode of travel (i.e., by car, on foot, by bicycle) in both trip directions (routes to and from school), without reporting the frequency. This dichotomous variable frequently occurs in the scientific literature, although no absolute consensus has yet been reached on dichotomizing active versus passive [20]. The number of active or passive journeys per week (i.e., 0 to 10 trips) seems important, not only to categorize adolescents in terms of active or passive commuting, but also to know how active or passive they are [21]. However, the continuous variable may be categorized as a dichotomous variable, although there is still no agreement on the cut-off criteria for dichotomizing active versus passive commuters.

Taking into account the number of active trips to define and differentiate active from passive commuters, there is a wide range of cut-off criteria across literature [17]. For example, while Bere and Bjørkelund [22] classified adolescents as active commuters when they actively commuted more than 50% of the times, Carver, Timperio and Crawford [23] considered adolescents who reported more than seven out of 10 trips (i.e., 70%) as active commuters. Ruiz-Ariza and colleagues [24] recently classified adolescents as inactive when they made less than five out of 10 active journeys per week, or at least five weekly active trips lasting for more than 15 min. However, other studies have suggested a four-out-of-five (i.e., 80%) active trip criterion to categorize adolescents as active commuters [25,26]. A recent study [27] has identified individuals who used active transport means (i.e., walking, cycling, scooters, or skateboards) on three or more school days per week (i.e., 60%) as active commuters. These findings are similar to those reported by Ross and colleague [28] and Veitch and colleague [29]. Finally, considering the potentially different number of trips to school that a student may make (e.g., different schedules), Mah and colleagues [30] classified students who actively commuted more than 90% of the time as active commuters.

This variability in criteria makes it complicated to make comparisons between studies. For instance, the 0% of passive commuters found by Bere and Bjørkelund [22] using a 50% active trip cut-off criterion cannot be fully compared to the 29.7% of passive commuters found by Te-Velde and colleagues [26] using an 80% active trip cut-off criterion. Therefore, it seems important to shed more light on the most accurate cut-off to correctly evaluate the prevalence of active and passive commuting modes among adolescents [19,31]. This may help not only to identify changes in the proportion of ACS, but also to clarify some current issues, such as the effectiveness of interventions focused on active commuting, considering potentially relevant factors (e.g., the distance from home to school) [20]. In this vein, previous studies have identified a threshold distance below which adolescents are more likely to commute actively to school. The threshold distance to ACS depends on sociodemographic and contextual variables [32].

To the best of our knowledge, no studies to date have focused on identifying active trip cut-off criteria to categorize individuals as active or passive commuters based on an exact number of weekly active trips. To bridge this gap, the first objective of this study was to explore whether the percentage of active commuters identified in a sample of adolescents would be significantly different depending on the cut-off criteria used. The second objective was to analyze whether the threshold distance may be a relevant variable to be considered when selecting cut-off criteria to categorize the sample into active and passive commuters.

It should be pointed out that this study was based on the most common international range of active trips per week for high school students, from zero to ten possible active trips (i.e., five weekdays and two trips [there and back] per week).

## 2. Materials and Methods

### 2.1. Participants and Procedure

An initial sample of 158 healthy students with no disability (12th–13th grade and 60.8% girls) from a high school in the city of Huesca (Spain) took part in this study. Huesca is a mid-sized city located in the north-east of Spain. It has 52,399 inhabitants and an urban area of 6.75 km^2^ (4.21 km^2^ not counting the industrial area), with a population density of 7762.8 inhabitants/km^2^. Written informed consent was required from both parents and adolescents, and their participation was entirely voluntary and confidential. A paper-and-pencil survey was administered by the research team to students in their classrooms, which lasted for approximately 30 min, without the presence of teachers. The survey was administered in a normal week of April 2017. After applying the two inclusion criteria (i.e., living at a distance of less than 4500 m from the school [considering the specific urban area], and completing all the items on the questionnaire), 140 of the eligible 158 students (*M* = 17.15 ± 0.63 years old; 60.7% girls) were included in the final study (i.e., 88.6% valid rate). Approval from the Ethics Committee for Clinical Research of Aragon (CEICA) was obtained (Ethic code: C1P117/0018). All procedures were performed in accordance with the relevant guidelines and regulations of the Ethics committee.

### 2.2. Assessment of Sociodemographic Characteristics

Participants self-reported their age, gender and home address.

### 2.3. Assessment of Mode and Frequency of Commuting to/from School

Participants completed a self-report questionnaire on their usual as well as on their latest weekly commuting patterns to and from school (Monday to Friday), respectively. Similarly to previous studies [33], the usual commuting mode to/from school was assessed by asking the following questions: ‘How do you usually commute from home to school?’ and ‘How do you usually commute from school to home?’. The answer options were walking, cycling, by car, motorcycle, or bus. Walking and cycling were categorized as usual active transport modes, whereas travelling by car, motorcycle or bus were categorized as usual passive transport modes. Accordingly, participants were considered to be categorized as (1) usual active commuters (i.e., having a usual active transport mode to and from school, or (2) usual passive commuters (i.e., less than two usual active transport modes). These questions had been validated in the Spanish population [34] and were the result of a systematic review of 158 studies from the scientific literature on the assessment of the usual mode of commuting, using questionnaires [17].

With respect to the weekly commuting patterns, the frequency of ACS was expressed as the number of active trips per week to and from school (range: 0–10). Adolescents self-reported all their weekly trips to and from school using a single item from a validated school travel questionnaire for each travel day^34^:“How do you commute to school on ‘Mondays’?” and “How do you commute from school on ‘Mondays’?”, repeating this question for every weekday. Each question for every day of the week about commuting to/from school was categorized as ‘active commuting’ (i.e., walking or cycling) or ‘passive commuting’ (i.e., by car, motorbike or bus). All weekly active trips (i.e., ‘active commuting’) were summed into a single variable ranging from 0 to 10. Based on this variable, subjects were classified as ‘active commuters’ according to six increasing weekly cut-off criteria: ≥five active trips, ≥six active trips, ≥seven active trips, ≥eight active trips, ≥nine active trips, and 10 active trips.

### 2.4. Assessment of Distance from Home to School

The objective measure of the distance to school was estimated using Google Maps, selecting the shortest route on foot option between the home address and the school’s main entrance. Adolescents’ home addresses and their respective school addresses were entered into Google Maps using the ‘get directions’ function and the walking distance between the two points was recorded in meters. Using Google Maps as a GPS mapping resource has been recommended in research studies [35,36,37] as a method to measure walking and cycling routes to school.

### 2.5. Assessment of Perceived Barriers to ACS

Adolescents completed the BATACE scale (Spanish acronym for *Barreras en el Transporte Activo al Centro Educativo* and translated as Barriers to Active Commuting to School [38]). This scale includes 18 items and measures two factors: environmental and safety barriers (e.g., ‘It is too far’), and planning and psychosocial barriers (e.g., ‘It is easier to go by car’). The scale was completed for walking and cycling barriers, separately. All items were assessed using a four-point Likert scale ranging from (1) strongly disagree to (4) strongly agree. This scale has shown adequate reliability and validity in past research studies with Spanish populations [37].

### 2.6. Statistical Analysis

All statistical analyses were conducted using the Statistical Package for Social Sciences (SPSS, version 21.0, IBM Corp. Released 2012. IBM SPSS Statistics for Windows, Version 21.0. Armonk, NY: IBM Corp.) and the level of significance was set at *p* < 0.05. Percentages of active commuters were calculated both for usual commuting mode and for the weekly frequency of commuting to school for every cut-off criterion. Cochran’s Q analyses were conducted to analyze differences in percentages of active commuters between different weekly cut-off criteria.

Participants’ perceived barriers to ACS were analyzed for each weekly cut-off criterion through a binary logistic regression analysis. First, univariate regression analyses were run to examine the independent association between the commuting modes according to the six different weekly cut-off criteria (as dependent variables) and perceived barriers for walking and cycling to school (as independent variables). Then, considering only significant univariate barriers (i.e., those that significantly predicted dependent variables), a series of multivariate binary logistic regression analyses were conducted.

Finally, the threshold distance for ACS was calculated using the receiver operating characteristic (ROC) curve analysis based on the usual commuting mode (i.e., usual active or passive commuter), and distance from home to school. The Younden Index was calculated [39] to identify the threshold distance that best distinguishes active from passive commuters. The area under the curve values of 0.90 is considered excellent, 0.80 to 0.89 good, 0.70 to 0.79 fair, and less than 0.70 poor [40]. The sample was split according to the threshold distance. Using the two resulting samples (i.e., who lived above and below the threshold distance), a series of Cochran’s Q tests were conducted again to analyze differences in percentages of active commuters between different weekly cut-off criteria.

## 3. Results

From the final sample of 140 adolescents (*M* = 17.15 ± 0.63 years old; 60.7% girls), 80% of the participants were ‘usual active commuters’. No differences were found by gender in either of the two usual commuting modes [to school: χ^2^(3, 140) = 4.04, *p* = 0.26; from school: χ^2^(3, 140) = 5.57, *p* = 0.14; usual active commuters: χ^2^(1, 140) = 0.749, *p* = 0.39].

As observed in Table 1 and based on the six different weekly cut-off criteria, percentages of active commuters ranged from 75% to 88.6% of the participants, showing a decrease in the percentage of active commuters as the number of active trips increased. Cochran’s Q analysis showed significant differences between percentages of the different cut-off criteria [Q(5) = 63.468, *p* < 0.00]. Pair-comparison of Cochran’s Q analysis revealed that no significant differences were found between the percentages from the ≥six, ≥seven, and ≥eight active trip cut-off criteria. Similarly, no significant differences were found between the percentages of active commuters from the ≥nine and ten active trip cut-off criteria either.

Table 2 shows the association between significant univariate barriers to ACS and the commuting mode for each weekly cut-off criterion. It should be noted that the planning and psychological walking barrier, ‘It is easier to go by car’, was significantly and negatively associated with ACS for the seven (β = −0.85; SE = 0.41, *p <* 0.05) and the eight (β = −0.86; SE = 0.38, *p <* 0.05) active trip cut-off criteria. The environmental and safety walking barrier, ‘It is too far’, was significantly and negatively associated with ACS for the eight (β = −0.72; SE = 0.36, *p* < 0.05), nine (β = −0.96; SE = 0.35, *p* < 0.05) and ten (β = −0.1.03; SE = 0.35, *p* < 0.01) active trip cut-off criteria. The dependent variable eight active trip weekly cut-off criterion had the greatest number of significant barriers. 

The ROC curve analysis showed a threshold distance of 1350 m (area under the curve: 0.809; *p <* 0.001). In total, 60.7% (i.e., *n =* 85) of the total sample lived below the threshold distance (see Figure 1). The percentages of active commuters for each weekly cut-off criterion ranged from 90.6% to 97.6% for participants who lived below the threshold distance and from 50.9% to 74.5% for those who lived above the threshold (see Table 3). With respect to the participants who lived below the threshold distance, Cochran’s Q analysis showed significant differences between percentages of the different cut-off criteria [Q(5) = 23.182, *p <* 0.00]. Pair-comparison of Cochran’s Q analysis revealed that no significant differences were found between the six, seven, eight, nine, and ten active trip cut-off criteria. With regard to adolescents who lived above the threshold, Cochran’s Q analysis reported significant differences between percentages of different cut-off criteria [Q(5) = 41.282, *p <* 0.00]. A pair-comparison of Cochran’s Q analysis revealed that no significant differences were found between the percentage of active commuters between the eight, nine, and ten active trips cut-off criteria.

## 4. Discussion

The present study aimed to propose the most valid and reliable criteria to correctly classify adolescents as active or passive commuters, in agreement with the frequency of active trips to school.

As previous research has shown [22,30,41], the current study provides different percentages of active commuters depending on the various cut-off criteria used. Percentages ranged from 75% (i.e., the strictest cut-off criterion) to 88.6% (i.e., the least strict cut-off criterion). The present study reported significant differences between the percentage of active commuters for the total sample when ≥five, ≥eight and 10 active trip cut-off criteria were applied. Based on these results, it seems that the most recommendable cut-off criteria for adolescents fluctuated between the ≥eight and ≥nine active trip cut-off criteria.

Considering the ≥eight active trip cut-off criterion as a reference, the percentage of active commuters in this research study (i.e., 77.9%) is more similar to other studies that used the same active trip cut-off criterion (i.e., 76.2% of active commuters [26]), although in those studies, data were collected from 11–12-year-old children. On the other hand, when the ≥eight active trip cut-off criterion is chosen, the percentage of adolescents classified as active commuters is much higher (i.e., 18.7%) in the current study, in contrast to previous research [30]. This controversy in the results may be explained by other variables that influence ACS, such as age [42], socio-cultural context [43], climate [44], socio-economic status [45], and city size [46].

A multivariate binary logistic regression analysis was conducted to corroborate which would be the most accurate criterion, including perceived barriers to ACS. A positive association between the number of perceived barriers and active commuting has been observed in previous studies [47]. Our results suggest that the ≥eight active trip cut-off criterion could be considered as the most accurate, because it proved to be associated with a greater number of planning and environmental barriers. In addition, these results are similar to those reported by Becker and collegues^47^. These authors found similar barriers in adolescents (e.g., ‘It is easier to go by car/bus’), which have been highlighted in the present research when applying the ≥eight active trip cut-off criterion. Moreover, an average of 7.7 active trips/week has recently been reported in a large sample of Spanish children and adolescents [48], which reinforces the choice of the ≥eight active trip cut-off criterion as the most suitable in this context. Nevertheless, the ≥nine active trip cut-off criterion should not be forgotten in future methodological studies.

A vast body of studies has suggested that there is a clear association between distance from home to school and the use of an active commuting mode, highlighting that participants living closer to school are more likely to actively commute than those who live further away [32,49,50,51]. Consequently, the most accurate cut-off criteria may hypothetically oscillate when the distance from home to school variable is considered in the analysis. Regarding adolescents who lived below the threshold distance (i.e., <1350 m from home to school), our results did not show any variation between the ≥six active trip cut-off criterion and the 10 active trip cut-off criterion in terms of the percentage of subjects classified as active commuters (i.e., 91.8%–90.6% of active commuters). Therefore, a significant oscillation of the percentage of active commuters was only found between the ≥five active trip cut-off criterion (i.e., 97.6% active commuters) and all the other cut-off criteria. Considering these results, the present study also provides evidence about the need for greater strictness in the selection of a cut-off criterion when the distance to school becomes shorter. Assuming that distance might not constitute a major barrier to ACS (i.e., living below the threshold distance), and given that ACS seems to be a stable behavior, some stricter cut-off criteria could be adopted to correctly differentiate active commuters from passive commuters in adolescence.

Turning our attention to adolescents who lived above the threshold distance (i.e., ≥1350 m from home to school), different percentages of active commuters were displayed, depending on the cut-off criterion chosen. This decision resulted in lower rates of active commuters as the criteria became stricter. Cochran’s Q analysis findings suggest that the most accurate cut-off criterion should be either the ≥seven or the ≥eight active trip cut-off criteria. Although the size of the city could be an influential variable in these results, distance from home to school should be considered an important variable (i.e., living above the threshold distance) when methodological recommendations are stated. Further research should focus on the influence of distance on ACS behavior.

In sum, this study suggests that the ≥eight active trip cut-off criterion could be considered as optimal for the current sample. It also indicates the need to consider other variables (e.g., distance) to apply this criterion in different contexts. The results of the research on ACS may be influenced by the correct choice of cut-off criteria. This has already occurred on previous occasions with the methodological decisions adopted to measure PA (e.g., epoch length in accelerometers; for further information [52]). Consequently, similar considerations should be taken into account when using frequency of active trips to school to categorize people as active or passive commuters. This fact will allow researchers to conduct rigorous, standardized and comparable ACS studies. This is promising, but more research is needed, however, to confirm our findings.

This work has several limitations. Firstly, our findings may not be generalizable to other contexts. This study was developed in a specific socio-cultural context using a convenience sample of a specific age. Therefore, further research should address different cultural contexts. Secondly, another limitation of this study is related to the self-report questionnaire used for data collection. Even though a children’s self-report questionnaire seems to be the most common way of assessing ACS [17], this method should be used with caution because it has been proven to agree poorly with objective estimates [31].

Despite these limitations, this study has notable strengths. To our knowledge, this is the first research that raises the issue of cut-off criteria to classify active commuters in adolescence according to the number of weekly active trips to school. Although it is only an approximation and more research is required, the current study begins to draw conclusions about the need to establish common criteria to compare international results. Nevertheless, the need to consider the characteristics of each context and other potential moderator variables in the analysis is also stated. These issues should be addressed in future research.

## 5. Conclusions

The literature has also shown some inconsistencies in the categorization of active and passive commuters. The heterogeneity of research questions and methods makes it difficult to compare prevalence between studies and to investigate the influence of ACS on health-related outcomes^17^. According to the objectives of this research, significant differences were found in the percentage of active commuters, depending on the cut-off criteria used. In addition, the results show the relevance of the threshold distance when selecting the most accurate cut-off criterion. Nonetheless, the intention of our study was not to conclude with any final and decisive recommendations about the ideal cut-off criteria to differentiate active from passive commuters. Our sole aim was to pose a cutting-edge question on this methodological topic that should be addressed by the scientific community in coming years. Although there is still no consensus regarding the optimal cut-off criteria for dichotomizing active versus passive commuters [17], we tried, to some extent, to provide some evidence to clarify this issue in the adolescent population.

Prior to designing any intervention to promote ACS, this methodological issue should be considered for any population, paying special attention, in particular, to the characteristics of the intervention context. We also truly encourage researchers to compare our recommendations with other proposals in order to find the best criteria.

## Figures and Tables

**Figure 1 ijerph-16-03796-f001:**
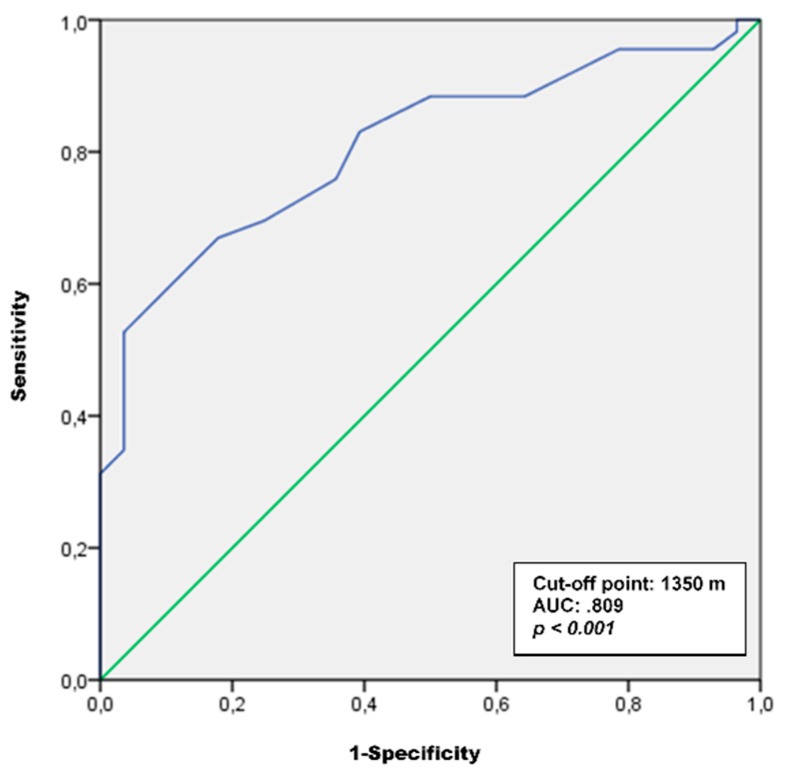
Receiver operating characteristic (ROC) curve analysis for active versus passive commuters by distance.

**Table 1 ijerph-16-03796-t001:** Cochran’s Q test values for the percentage of active commuters depending on the number of weekly active trips to and from school.

*n* = 140	1	2	3	4	5	6	%
1. ≥five active trips	-	13.00 **	14.00 **	15.00 **	18.00 **	19.00 **	88.6
2. ≥six active trips		-	1.00	2.00	5.00 *	6.00 *	79.3
3. ≥seven active trips			-	1.00	4.00 *	5.00 *	78.6
4. ≥eight active trips				-	3.00	4.00 *	77.9
5. ≥nine active trips					-	1.00	75.7
6. =10 active trips						-	75.0

Note: * *p* < 0.05, ** *p* < 0.01. % = percentage of active commuters.

**Table 2 ijerph-16-03796-t002:** Multivariate binary logistic regression analysis of different active trip cut-off criteria based on children’s active commuting to school (ACS) barriers.

Variables	≥Five Active Trips			≥Six Active Trips			≥Seven Active Trips			≥Eight Active Trips			≥Nine Active Trips			=10 Active Trips		
	β	SE	*p*	β	SE	*p*	β	SE	*p*	β	SE	*p*	β	SE	*p*	β	SE	*p*
**Walking barriers (*n =* 108)**																		
Planning and psychosocial barriers																		
I get hot and sweat, or it is always raining													−0.28	0.36	0.43			
It is easier to go by car	−0.81	0.52	12	−0.67	0.37	0.07	**−0.85**	**0.41**	**0.04**	**−0.86**	**0.38**	**0.02**	−0.55	0.32	0.09	−0.62	0.33	0.06
It involves too much planning	−0.38	0.43	0.37	−0.28	0.36	0.43	−0.20	0.36	0.60	−0.40	0.36	0.28	−0.07	0.34	0.83	−0.31	0.35	0.36
I do not enjoy myself							−0.77	0.47	0.11									
Environmental and safety barriers																		
It is too far	−0.75	0.50	0.13	−0.67	0.36	0.06	−0.65	0.38	0.09	**−0.72**	**0.36**	**0.04**	**−0.96**	**0.35**	**0.01**	**−10.03**	**0.35**	**0.00**

Note: Only significant univariate barriers were included in the analysis of each dependent variable. SE = Standard Error. Bold fond = *p* < 0.05.

**Table 3 ijerph-16-03796-t003:** Cochran’s Q test values for the percentage of active commuters depending on the number of weekly active trips to and from school for participants below and above the threshold distance.

Cut-off Criteria	1	2	3	4	5	6
1. ≥five active trips	-	8.00 **	9.00 **	10.00 **	12.00 **	13.00 **
2. ≥six active trips	5.00 *	-	1.00	2.00	4.00 *	5.00 *
3. ≥seven active trips	5.00 *	0.00	-	1.00	3.00	4.00 *
4. ≥eight active trips	5.00 *	0.00	0.00	-	2.00	3.00
5. ≥nine active trips	6.00 *	1.00	1.00	1.00	-	1.00
6. =10 active trips	6.00 *	1.00	1.00	1.00	0.00	-
% of active commuters above the threshold	74.5	60.0	58.2	56.4	52.7	50.9
% of active commuters below the threshold	97.6	91.8	91.8	91.8	90.6	90.6

Note: * *p <* 0.05, ** *p <* 0.01; Q-Cochran test values from students who lived below the threshold distance (*n =* 85) are displayed below the diagonal, and signification levels from students who lived above the threshold distance (*n =* 55) are displayed above the diagonal.

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
