# Peer review of "Active or Passive Commuter? Discrepancies in Cut-off Criteria among Adolescents"

_ijerph, 2019, doi:10.3390/ijerph16203796_

Round 1

Reviewer 1 Report

The study raise an important issue in the field of active transport to school studies. The differences between estimations of active commuters found in the literature is a good starting point for the analysis. Moreover, the authors also provide a good insight about the relation of the threshold distance to school and the cut-off points used to classified subjects in active or passive commuters.

The manuscript is well organized and deserves publication. The only point I would like to rise is the discussion about the possible use of the cut pints more than 7 or more than 8 trips that was briefly pointed in line #240.

I would like to thank authors and editors for the opportunity to read and contribute to the publication of this important manuscript.

Author Response

Point 1. English review

Response 1. The manuscript has been reviewed and edited by a professional native English translator. Please find attached a certificate. 

Point 2. The manuscript is well organized and deserves publication. The only point I would like to rise is the discussion about the possible use of the cut pints more than 7 or more than 8 trips that was briefly pointed in line #240.

Response 2. The discussion has been updated according to your suggestion. You can check it among lines #234-237.

Do not hesitate to contact me for any further requirements.

Reviewer 2 Report

The aim of this study was to explore whether the percentage ofactive commuters identified in a sample of adolescents is significantly different according to the cut-off criteria used, and to analyze whether the threshold distance may be a relevant variable to consider when selecting cut-off criteria to categorize the sample into active and passive commuters assess the association between coffee intake and self-reported. The paper is very great and has an audience good for readers. However, looking for improve the paper, few issues are addressed.

-In the abstract is crucial to add more results.

-Introduction is too long.

-How the Assessment of sociodemographic characteristics were done?

-What is the sample size?

-The conclusion did not answer the aim of study. Please, rewritten.

-What is exclusion criteria?

-Please, add anthropometric data.

-Who is these adolescents? Healthy? Obese?

-Please, to add the ROC curve in the manuscript.

-An extensive editing of English language and style are required.

Author Response

Please find attached a new version of the manuscript including a review on the basis of your valuable suggestions.  

Point 1. In the abstract is crucial to add more results.

Response 1. The abstract has been updated according to your suggestion. You can check it between lines # 17-19

Point 2. Introduction is too long.

Response 2. The introduction has been updated according to your suggestion. You can check it between lines #30-31

Point 3. How the Assessment of sociodemographic characteristics were done?

Response 3. The Assessment of sociodemographic characteristics has been updated according to your suggestion. You can check it in line #104

Point 4. What is the sample size?

Response 4. The sample size is described in line #99

Point 5. The conclusion did not answer the aim of study. Please, rewritten.

Response 5. The conclusion has been updated according to your suggestion. You can check it between lines # 289-291.

Point 6. What is exclusion criteria?

Response 6. Inclusion criteria have been adressed in "Material and methods" (lines #97-100). We consider that exclusion criteria are explained as failing to meet aforementioned inclusion criteria.

Point 7. Please, add anthropometric data.

Response 7. Anthropometric data have not been measured because they do not answer the aim of this research.

Point 8. Who is these adolescents? Healthy? Obese?

Response 8. Participants belonged to a natural group, from one high school in the city of Huesca (as explained in "Materials and methods"), resulting in a convenience sample. As a result, we haven't looked into variables as health status and BMI, because they are not relevent to the objective purseued. 

Point 9. Please, to add the ROC curve in the manuscript.

Response 9. All parameters needed to understand the ROC curve analysis have been reported in "Results" section (lines # - #). We decided not to include the figure as a way to avoid redundant information.

Point 10. An extensive editing of English language and style are required.

Response 10. The manuscript has been reviewed and edited by a professional native English translator. Please find attached a certificate. 

Do not hesitate to contact me for any further requirements.

Round 2

Reviewer 2 Report

-The sample size calculus is not clear. 

-Who is these adolescents? Healthy? Obese? This information is crucial to explain the population. These informations may help to replicate the data in future.

-The inclusion of ROC curve is better to understand the data.

Author Response

Point 1. The sample size calculus is not clear. 

Response 1. You can see the changes between lines #98-100

Point 2. Who is these adolescents? Healthy? Obese? This information is crucial to explain the population. These informations may help to replicate the data in future.

Response 2. You can see the changes in line #90

Point 3. The inclusion of ROC curve is better to understand the data.

Response 3. We include the ROC curve as figure 1 in #line 204

Thank you for your suggestions.